# Sturgeon Parasites: A Review of Their Diversity and Distribution

György Deák , Elena Holban, Isabela Sadîca and Abdulhusein Jawdhari *

National Institute for Research and Development in Environmental Protection, 294 Splaiul Independenței Str., 060031 Bucharest, Romania; dkrcontrol@yahoo.com (G.D.); holban.elena@yahoo.com (E.H.); isabelasadica01@gmail.com (I.S.)
* Correspondence: h.jawdhari20@s.bio.unibuc.ro

**Abstract:** Sturgeon species have inhabited the world's seas and rivers for more than 200 million years and hold significant taxonomic significance, representing a strong conservation interest in aquatic biodiversity as well as in the economic sector, as their meat and eggs (caviar) are highly valuable goods. Currently, sturgeon products and byproducts can be legally obtained from aquaculture as a sustainable source. Intensive farming practices are accompanied by parasitic infestations, while several groups of parasites have a significant impact on both wild and farmed sturgeons. The present article is a review of common sturgeon parasites from the genus: Protozoa, Trematoda, Crustacea, Nematodes, Monogenea, Hirudinea, Copepoda, Acanthocephala, Cestoda, Polypodiozoa, and Hyperoartia, while also addressing their pathology and statistical distribution.

**Keywords:** sturgeons; parasites; pathology; statistical distribution; freshwater

## 1. Introduction

Sturgeons are members of the *Acipenseriformes* order, which is over 200 million years old and comprises twenty-seven species and two families, the *Acipenseridae* and *Poyodontidae* [1]. The species has a long life cycle and is native to the Northern Hemisphere [2,3]. The natural habitats of sturgeons are the freshwaters of Europe, Asia and North America. The species inhabit inland water, bays, estuaries, and the coastal regions of seas and oceans. Although most sturgeon species migrate and spawn in freshwater, they also spend a significant amount of their life cycle in brackish water. The Caspian basin accounted for up to 90% of caught sturgeon, but as many fisheries have collapsed, many individual species or populations are now endangered. Two such examples are the *Huso* and *Acipenser* genera, with a total of 17 species [4], the most well-known of which are *Acipenser ruthenus*, *Acipenser stellatus*, *Acipenser gueldenstaedti*, *Acipenser oxyrhynchus*, *Huso huso*, *Acipenser persicus*, *Acipenser sturio*, and *Acipenser naccarii*. Sturgeon species differ significantly from other fish species not solely because of their anatomy, but also because of their longevity and behavior. The Danube is a well-known habitat for sturgeon populations, although in recent years only four species were found [5] out of the six previously known [6]. Such decreases in sturgeon populations are mainly due to dams, pollution, and overfishing [7,8]. With the effective wild population decreasing, the sturgeon parasitic infestations are becoming more important and, like in many other fish species, they are becoming more prominent as aquaculture is expanding [9].

Sturgeons are host to many parasites, such as protozoans, trematodes, nematodes, monogeneans, helminths, and argulidaes [10]. These parasites are among the most significant factors responsible for weight loss, impotence, strange behaviour, deformed gills, and epithelial lesions [11], ultimately resulting in the diminishing of wild stocks or in financial losses in the case of fish farming. In addition, external parasites have the potential to spread bacteria, viruses, and other pathogens, resulting in a boost of secondary fungal, bacterial, and viral infections [12,13]. Sturgeons are susceptible to viral, parasitic, and bacterial

infections, fungal infections being generally rare [14]. Parasites and parasite communities are important environmental quality bioindicators because they are part of the aquatic biodiversity and are influenced by it, either directly or indirectly, through their hosts, despite the fact that they are frequently ignored as a biological component for ecological assessment [15]. Therefore, environmental monitoring programs are more effective when parasites and parasitic fish communities are also regarded [16].

Protozoan parasites are a diverse group of single-celled organisms that can reside either inside or on the surface of their host. They do not always cause disease in fish, but they may be present in a subclinical or carrier state [17,18]. Certain species of ciliate protozoa are known to consume bacteria and other microorganisms that may cause infections in fish. By controlling the population of potentially harmful microorganisms, these protozoa may contribute to supporting fish health. On the other hand, some protozoa can be harmful to sturgeon, causing disease and other health problems, for example, *Ichthyophthirius multifiliis*, which can infect sturgeon as well as other fish, causing the white spot disease that causes the skin and gills of these species to be susceptible to other parasites during different stages of rearing, thereby severely affecting their normal growth and development [12]. Due to their varied life cycles, which include time spent in freshwater, brackish water, and marine water, sturgeon in the wild are more disease-resistant than other fish species [19] but still subject to parasitic infestation. The current review documents the presence of both ecto- and endoparasitic species that occur on sturgeon in their natural environments. A comprehensive list of sturgeon parasites and organs affected is presented in Table 1.

**Table 1.** Parasitic species (ecto- and endoparasites species) of sturgeons and the infected organs.

| Sturgeon Species | Parasitic Species | Infected Organs | Reference |
|---|---|---|---|
| *Acipenser stellatus* (Pallas, 1771) | Protozoa *Ichthyophthirius multifiliis* | G. S. F | [20] |
| *Acipenser oxyrhynchus* (Mitchill, 1815) | Protozoa *Trichodina* sp., *Apiosoma* sp., Monogenea *Gyrodactylus* sp. Crustacea *Ergasilus siebold*, *Argulus coregoni* | G. S. F | [21] |
| *Acipenser persicus* (Borodin, 1897) *Acipenser guldenstadti* (Brandt & Ratzeburg, 1833) *Acipenser stellatus* | Protozoa *Cryptobia acipenseris*, *Haemogregarina acipenseris* | BL | [22] |
| *Acipenser oxyrinchus* | Protozoa *Chilodonella* sp. | S. D | [23] |
| *Acipenser gueldenstaedti and Acipenser baerii* (Brandt, 1869) | Protozoa *Trichodina reticulate*, Trematoda *Diplostomum spathaceum* | G. S. N | [24] |
| *Acipenser persicus* (Borodin, 1897) | Protozoa *Trichodina* sp., *Ichthyophthirius multifiliis* Trematoda *Diplostomum spathaceum* Nematoda *Cucullanus sphaerocephlaus*, *Anisakis* sp., *Skyrjabinopsilus semiarmatus* and *Leptorhynchoides plagicephalus* | S. F. G. I | [25] |
| *Acipenser persicus* | Protozoa *Trichodina reticulate* Trematoda *Diplostomum spathaceum* | S. F. G. E | [26] |
| *Acipenser oxyrinchus* | Monogenea *Nitzschia sturionis* | S. G | [27] |
| *Acipenser gueldenstaedtii Acipenser persicus Acipenser stellatus*, *Acipenser sturio* (Linnaeus, 1758) | Protozoa *Ichthyobodo necatrix*, *Trichodinidae*, *Apiosoma*, *Epistylis* Monogeneans *Diclybothrium*, *Dactylogyrus* and Crustaceans *Agulus foliaceus* | G. S. F | [19] |
| *Acipenser ruthenus* (Linnaeus, 1758) | Protozoa *Ichthyophthirius multifiliis* Trematoda *Skrjabinopsolus semiarmatus*, *Acrolichanus auriculatus* | S. GU | [28] |
| *Acipenser persicus* | Protozoan *Ichthyophthirius multifiliis* | G | [29] |

**Table 1.** *Cont.*

| Sturgeon Species | Parasitic Species | Infected Organs | Reference |
|---|---|---|---|
| *Acipenser ruthenus* | Protozoa *Cryptobia acipenseris, Haemogregarina acipenseris, Trichodina* sp. Cestoda *Proteocephalus* sp. Trematoda *Crepidostomum auriculatum, Diplostomum chromatophorum,* Nematoda *Capillospirura ovotrichuria,* Acanthocephala *Echinorhynchus cinctulus,* Hirudinea *Piscicola geometra,* Copepoda *Ergasilus sieboldi* | BL. F. I. | [30] |
| *Acipenseridae* | Trematoda *Skrjabinopsolus semiarmatus, Sanguinicola Posthodiplostomum,* Cestoda *Amphilina foliacea* Nematoda *Contracaecum* sp., *Acanthocephala, Pomphorhynchus bosniacus.* | I | [11] |
| *Acipenser fulvescens* (Rafinesque, 1817) | Trematoda *Pristicola bruchi* | I | [31] |
| *Acipenser fulvescens* | Trematoda *Acipensericola glacialis* | H | [32] |
| *Acipenser persicus* | Trematode *Skrjabinopsolus semiarmatus* Nematodes *Cucullanus sphaerocephalus, Eustrongylides, Anisakis* sp. cestode *Amphilina foliacea* Monogenea *Diclybothrium armatum, Nitzschia storionis* Acanthocephalan *Leptorhynchoides plagicephalus* Crustacea *Pseudotracheliastes stellatus* | G. S. F. I | [33] |
| *Acipenser ruthenus* | Trematoda *Skrjabinopsolus semiarmatus* | Ds | [34] |
| *Acipenser schrenkii* (Brandt, 1869) | Trematoda *Crepidostomum oschmarini* | I | [35] |
| *Acipenser sturio, Acipenser ruthenus, Acipenser fulvescens* | Trematoda *Skrjabinopsolus semiarmatus, Distomum hispidum, Deropristis hispida, Cestrahelmins rivularis, Homalometron armatum.* | I | [36] |
| *Acipenser oxyrinchus* | Trematoda *Skrjabinopsolus nudidorsalis* sp. | I | [37] |
| *Acipenser stellatus Acipenser gueldenstaedtii Acipenser nudiventris* (Lovetsky, 1828) *Acipenser Huso huso dauricus* (Georgi, 1775) | Trematode *Skrjabinopsolus semiarmatus* Nematode *Cucullanus sphaerocephalus, Eustrongylides excisus* Cestodes *Amphilina foliacea, Bothrimonus fallax* Acanthocephalan *Leptorhynchoides plagicephalus* | I | [38] |
| *Acipenser persicus* | Trematoda *Skrjabinopsolus.* Nematode *Cucullanus sphaerocephalus, Eustrongylides excisus* Cestodes *Amphilina foliacea, Bothrimonus fallax* Acanthocephalan *Leptorhynchoides plagicephalus* | I | [39] |
| *Acipenser oxyrinchus* | Copepoda *Dichelesthium oblongum* Monogenea *Nitzschia* sp. | G. F. | [40] |
| *Acipenser oxyrinchus* | Copepoda *Dichelesthium oblongum* | S | [41] |
| *Husu huso* | Copepoda *Argulus* | G. S. F | [42] |
| *Acipenser oxyrinchus* | Copepoda *Argulus flavescens* | G | [43] |
| *Acipenser oxyrinchus* | Copepoda *Dichelesthium oblongum* | G | [44] |
| *Huso huso, Acipenser ruthenus, Acipenser gueldenstaedtii* | Copepoda *Lernaea cyprinacea, Argulus* sp., *Ergasilus* sp. | G. S. F | [45] |
| *Acipenseriformes* | Copepoda *Tracheliastes gigas* | B | [46] |
| *Acipenser fulvescens* | Copepoda *Argulus* sp. | G. S | [47] |
| *Acipenser oxyrinchus* | Copepoda *Dichelesthiidae oblongum* | G | [48] |
| *Acipenser oxyrinchus* | Copepoda *Caligus elongatus, Dichelesthium Oblongum* Hirudinea *Calliobdella vivida* Crustacea *Argulus stizostethii* and Monogenea *Nitzschia sturionis* | F. S. B | [49] |
| *Acipenser ruthenus* | Polypodiozoa *Polypodium hydriforme* | Eg | [50] |

Table 1. *Cont.*

| Sturgeon Species | Parasitic Species | Infected Organs | Reference |
|---|---|---|---|
| *Acipenser ruthenus* | Polypodiozoa *Polypodium hydriforme* | Eg | [51] |
| *Acipenser mikadoi* (Hilgendorf, 1892) | Polypodiozoa *Polypodium hydriforme* Cestoda *Amphilina japonica* Hirudinea *Limnotrachelobdella* sp. | Eg. B. I | [52] |
| *Acipenser fulvescens* | Polypodiozoa *Polypodium hydriforme* | Eg | [51] |
| *Acipenser fulvescens* | Polypodiozoa *Polypodium hydriforme* | Eg | [53] |
| *Acipenseriform* | Polypodiozoa *Polypodium hydriforme* | Eg | [54] |
| *Acipenser fulvescens* | Polypodiozoa *Polypodium hydriforme* | Eg | [55] |
| *Acipenser fulvescens* | Polypodiozoa *Polypodium hydriforme* | Eg | [56] |
| *Acipenser, Huso dauricus, Acipenser schrenckii* (Brandt, 1869) | Polypodiozoa *Polypodium hydriforme* | Eg | [57] |
| *Acipenser mikadoi* | Polypodiozoa *Polypodium hydriforme* Hirudinea *Limnotrachelobdella* | Eg. B | [58] |
| *Acipenseriformes* | Polypodiozoa *Polypodium hydriforme* | Eg | [59] |
| *Acipenser gueldenstaedtii* | Hirudinea *Acipenserobdella volgensis* | F | [60] |
| *Acipenser oxyrinchus* | Hirudinea *Caspiobdella fadejewi* | B | [61] |
| *Acipenser brevirostruin* (Lesueur, 1818) | Hirudinea *Placobdella montifera, Piscicola geometra* | B | [62] |
| *Acipenser oxyrinchus* | Hirudinea *Calliobdella vivida* | G | [63] |
| *Acipenser baerii, Acipenser ruthenus* | Nematoda *Raphidascaris acus* | I | [64] |
| | Cestodes *Amphilina foliacea, Bothrimonus fallax,* Nematoda *Cucullanus sphaerocephalus, Leptorhynchoides plagicephalus* and Acanthocephalan *Eustrongylides excisus* | I | [65] |
| *Acipenser ruthenus* | Nematoda *Cystidicoloides ephemeridarum* | I | [66] |
| *Acipenser stellatus* | Nematoda *Eustrongylides excisus* | Gu | [67] |
| *Acipenser persicus* | Nematoda *Cucullanus sphaerocephalus,* Trematode *Skrjabinopsolus semiarmatus,* Cestoda *Eubothrium acipenserinum.* Acanthocephala *Leptorhynchoides plagicephalus* | I | [26] |
| *Acipenser fulvescens* | Nematoda *Capillospirura* sp. | I | [68] |
| *Acipenser filvescens* | Nematoda *Cucullanus sphaerocephala* Trematoda *Skrjabinopsolus semiarmatus* Acanthocephala *Leptorhynchoides plagicephalus* Cestoda *Amphilina foliacea* | I | [69] |
| *Acipenser transmontanus* (Richardson, 1836) | Nematoda *Cystoopsis acipenseri* | I | [70] |
| *Acipenser fulvescens* | Monogenea *Diclybothrium atriatum* Trematoda *Skrjabinopsolus semiarmatus* | G. S. I | [71] |
| *Acipenser persicus Acipenser stellatus Acipenser gueldenstaedti Acipenser nudiventris* | Monogenea *Nitzschia sturionis, Diclybothrium* | G. I | [72] |
| *Acipenser baerii* | Monogenea *Diclybothrium armatum* | G | [73] |
| *Acipenser stellatus* | Monogenea *Nitzschia sturionis* | G | [74] |
| *Acipenser nudiventris* | Acanthocephala *Leptorhynchoides polycristatus* | I | [75] |
| *Acipenser naccarii* (Bonaparte, 1836) | Acanthocephala *Leptorhynchoides plagicephalus* | T | [76] |
| *Acipenser nudiventris* | Cestoda *Bothrimonus fallax* | I | [77] |
| *Acipenser stellatus* | Cestoda *Amphilina foliacea* | I | [78] |

| Sturgeon Species | Parasitic Species | Infected Organs | Reference |
|---|---|---|---|
| *Acipenser ruthenus* | Cestoda *Amphilina foliacea* | I | [79] |
| *Acipenser gueldenstadti* | Cestoda *Bothrimonus fallax, Eubothrium acipenserinum* | I | [80] |
| *Acipenser fulvescens* | *Hyperoartia* (*Lamprey*) *Petromyzon marinus* | B | [81] |
| *Acipenser fulvescens* | *Hyperoartia* (*Lamprey*) *Petromyzon marinus* | B | [82] |
| *Acipenser fulvescens* | *Hyperoartia* (*Lamprey*) *Petromyzon marinus* | B | [83] |
| *Acipenser fulvescens* | *Hyperoartia* (*Lamprey*) *Petromyzon marinus* | B | [84] |
| *Acipenser transmontanus* (Richardson, 1836) | Nematoda *Cystoopsis acipenseri* | I | [85] |
| *Acipenser transmontanus* | Trematoda *Crepidostomum auriculatum* Cestoda *Diphyllobothrium* sp, *Amphilina bipunctata.* Nematoda *Anisakis simplex.* Acanthocephala *Corynosoma strumosum* | I | [86] |
| *Acipenser transmontanus* | Allocreadiidae *Crepidostomum auriculatum* Monogenea *Nitzschia quadritestes* sp. Cestoda *Amphilina foliacea* | I | [87] |

G gill, S skin, F fin, I intestine, E eye, SP Spleen, H heart, N nose, Eg eggs, B body, BL blood T testes, GU gut, Ga gastrointestinal, Ds digestive system.

Various types of texts, such as abstracts, reviews, and original research articles, were examined and evaluated individually to determine their eligibility based on specific criteria. These included recording information about the sturgeon species, the parasite species, the site of infection, and the country of origin.

## 2. Sturgeons and Parasites

### 2.1. Protozoa, Monogenea and Crustaceans

Protozoa comprise a diverse group of predominantly single-celled eukaryotic organisms [75]. Depending on their species, protozoa can be either ectoparasites or endoparasites. Among cultured fish, ectoparasitic protozoa are the most commonly encountered parasites [88]. These parasites induce a reactive hyperplasia of the fish epithelium, and excessive mucus infestation can lead to gill hyperplasia, including epithelial hyperplasia of the entire gill filament, inflammation, hemorrhage, and necrosis [89]. Protozoa pose a significant threat to fish health, causing diseases in both farmed and wild populations [90]. Within fish populations, parasitic protozoa can rapidly spread, particularly those with direct life cycles and broad host specificity [91]. Some protozoa act as ectoparasites, residing on the skin, fins, and gills, while others invade internal organs, such as the intestine [92]. Parasite invasion can impede fish development, cause weight loss, and disrupt reproductive processes. In severe cases, infections can lead to long-term mortality and substantial damage to fish populations [93].

In a study conducted by Vasile et al. (2019) [20], the protozoan parasite *Ichthyophthirius multifiliis* was identified in *Acipenser stellatus* at a research hatchery in Romania. *Ichthyophthirius multifiliis* is a parasitic ciliate that was initially described by the French parasitologist Fouquet in 1876. This parasite has the ability to infect a wide range of freshwater fish species, including sturgeon. Upon infecting sturgeon, the parasite attaches itself to the skin and gills, where it feeds on bodily fluids, resulting in the formation of visible white spots on the outer layer of the fish, commonly referred to as "white spot disease".

*Ichthyophthirius multifiliis* can cause significant harm to the fish, including irritation, inflammation, tissue damage, reduced growth, weakened immune function, and even death [94,95]. The disease caused by this tissue-feeding parasite is referred to as ichthyophthiriasis [96]. Outbreaks of *I. multifiliis* occur when conditions are favorable for rapid multiplication, making it a major concern in aquaculture settings [97].

In another study conducted by Popielarczyk and Kolman (2013) [21] with *Acipenser oxyrinchus oxyrinchus* specimens obtained from an open system pond in Kuźniczka, Poland, protozoa, monogenean, and crustacean parasites were identified. The protozoa *Trichodina* sp. and *Apiosoma* sp., as well as the monogenean *Gyrodactylus* sp. and the crustaceans *Ergasilus sieboldi* and *Argulus coregoni*, were observed in specimens from various water habitats. A high number of *Trichodina* sp. parasites with varying morphology and size were found. These parasites resemble caps and range in size from 18 to 50 μm, with some specimens measuring up to 80 μm. They possess partial ciliation and have a saucer-shaped body, which enables them to move along the skin, fins, and gills of fish. *Trichodina* sp. parasites reproduce through binary fission; after reproduction they can either reattach to the same host or seek a new host in the water column.

The parasite possesses cilia arrangements on its body, which include a distinctive ring of denticles, morphological features that are crucial for species identification. Additionally, this parasite is classified as an ectoparasite that exhibits rapid movement on the gills, fins, and body surface of its host (in certain species, it can even inhabit the urinary tract). The *Trichodinidae* family, to which this parasite belongs, is known to cause trichodinosis [98] (also known as trichodinads), characterized by hyperplasia of the epithelium [99].

*Apiosoma* sp. parasites are not typically considered highly dangerous, but they can still inflict damage on sturgeons by attaching to their fins, gills, or skin surface, which leads to the destruction of the epithelial tissue and impairment of organ function. These ciliates are ectocommensals, living on the gills and body surface of aquatic organisms, particularly the fry of freshwater fish [100]. They are large, bell-shaped organisms measuring approximately 50–70 μm in length and 18–40 μm in width. The species have a free-living lifestyle but frequently attach themselves to various organisms in the water, including fish [98]. When sturgeons become infected with *Trichodina* sp. and *Apiosoma* sp., mucus accumulates on the skin surface, especially around the pectoral fin, gills, and gastrointestinal tract, as well as the oviduct and urinary bladder [96].

In aquaculture-related studies conducted in Romania, a total of 22 species of *Gyrodactylus* sp. within the monogenea group have been identified [101]. However, it should be noted that some sources report 145 species for the *Gyrodactylus* genus [102], while others mention up to 400 species [103]. The exact number remains uncertain due to synonyms and variations in taxonomic interpretation [103]. *Gyrodactylus* sp. is an ectoparasitic flatworm with a length of less than 1 mm and a body width of approximately 0.1 mm. It is characterized by a four-lobed head and an opisthaptor, which includes one prominent pair of large hooks and up to 12 smaller hooks. Infections caused by *Gyrodactylus* sp. can result in skin irritation and tissue damage. Additionally, in a study conducted by Choudhury (1997) [71], the monogenean species *D. atriatum* and the trematode *Skrjabinopsolus* sp. were identified in the gill of *Acipenser fulvescens* in Canada.

The copepod crustacea *Ergasilus sieboldi*, commonly known as "fish lice" [104], has been reported to infect sturgeons [21]. This parasite can have detrimental effects on the gills, as it attaches to the skin or gills and can cause physical damage, potentially leading to suffocation. Pazooki and Msoumian (2018) [22] identified the protozoa *Haemogregarina acipenseris* and *Cryptobia acipenseris* in *Acipenser persicus* and *Acipenser guldenstadti* in the southern part of the Caspian Sea. *Haemogregarina acipenseris* has an oval body shape, measuring 6.5–8.2 × 2.2–3.0 μm, with two rounded ends or one rounded and one sharpened end. The nucleus typically consists of a few chromatin granules, and it is commonly found in erythrocytes. *Haemogregarina acipenseris* has been previously recorded in sturgeons in the Caspian and Black seas, and it has been found in sterlet in the Volga and Danube rivers [30]. *Cryptobia acipenseris* is a parasitic protozoan measuring 11–16.4 μm in size [105]. The vegetative and sexual stages of this protozoan have been found in the blood of various sturgeon species [22]. While the majority of *Cryptobia acipenseris* live in the host's blood, some can also be found in the intestines and gills. Infections caused by *Haemogregarina acipenseris* and *Cryptobia acipenseris* can lead to severe consequences, including anemia and, finally, death [106]. Mohler et al. (2000) [23] identified the protozoan *Chilodonella* sp. in

*Acipenser oxyrinchus oxyrinchus* in the eastern side of Esopus Island in the Hudson River, New York. *Chilodonella* sp. is a single-celled organism belonging to the ciliate class of Alveolata. It is covered in cilia and possesses a dual nuclear structure. *Chilodonella* sp. is the causative agent of Chilodonelloza, a disease that affects the gills and skin of freshwater fish [107].

In the study conducted by Kayiş et al. (2017) [24], the protozoan *Trichodina reticulata* was identified in *Acipenser gueldenstaedtii* and *Acipenser baerii* in the Black Sea region of Turkey. This parasite infects the gill, skin, and fins, as mentioned in the previous reference [21]. Furthermore, Baska (1999) [28] discovered the presence of the protozoan *Ichthyophthirius multifiliis* and the nematode *Goussia acipenseris* in *Acipenser ruthenus* specimens in Hungary. Dobson and May (1987) [74] identified the monogenea *Nitzschia sturionis* in *Acipenser stellatus* specimens in the USA. Chebanov et al. (2013) [19] identified various protozoans, including *Ichthyobodo necatrix*, *Trichodinidae*, *Apiosoma* sp., and *Epistylis* sp., as well as the monogeneans *Diclybothrium* and *Dactylogyrus*, and the crustacean *Argulus foliaceus* in *Acipenser gueldenstaedti*, *Acipenser persicus*, and *Acipenser stellatus* in Russia. In Iran, Rahmati et al. (2021) [73] identified the monogenean *Diclybothrium armatum* in the gills of *Acipenser baerii*, and Matsche et al. (2010) [27] identified the protozoa *Nitzschia sturionis* in *Acipenser oxyrinchus oxyrinchus*.

## 2.2. Cestode, Trematode and Nematode

Fish-borne cestodes that can infect humans primarily belong to the order *Diphyllobothriidea* and are commonly referred to as broad tapeworms. These tapeworms have a three-host life cycle, with teleost fishes (excluding spirometra) serving as the second intermediate hosts and a source of human infection [27]. The larval form of the cestode penetrates the tissue of a crustacean host and undergoes metamorphosis into a proceroid. The fish becomes infected by consuming the crustacean. Once inside the fish, the adult cestodes gradually migrate to body organs and intestines, causing diseases and reducing the fish's lifespan [79].

The trematode *Diplostomum spathaceum* is responsible for a disease called diplostomatosis, or eye fluke disease [108]. This parasite has been found to be widespread among fish in Utah [109]. *Diplostomum spathaceum* has a complex life cycle involving multiple hosts and can cause significant harm. It attaches to the eye tissue of the fish and feeds on blood, leading to inflammation, swelling, and the formation of dark spots on the eye's surface [110]. Choudhury (2009) [31] identified the trematoda *Pristicola bruchi* in *Acipenser fulvescens* in Wisconsin, USA. *Pristicola bruchi* is smaller in size (ranging from 1.660 to 2.110 μm) than *Pristicola sturionis*. It possesses a single row of prominent peg-like oral spines instead of two rows, and its vitelline follicles dorsally converge over a small region without extending beyond the posterior testes. This is the first recorded occurrence of this genus in North America and appears to be the first report of the genus in sturgeon since the description of *Pristicola sturionis* in 1930 [111]. Warren et al. (2017) [32] identified the trematoda *Acipensericola glacialis* in *Acipenser fulvescens* during a survey of the Great Lakes Basin, specifically the Lake Winnebago system in the USA. *Acipensericola glacialis* derives its name from the Latin-specific epithet "glacialis" (glacier). The impact of this parasite on sturgeon can vary depending on the severity of the infestation and the overall health of the fish.

Nematodes and trematodes were identified in sturgeon by Sattari et al. (2006) in *Acipenser Persicus* (Persian sturgeon) in the southwest of the Caspian Sea, off the Guilan province of Iran [33]. The nematodes *Cucullanus sphaerocephalus*, *Eustrongylides excisus*, and *Anisakis* sp. were identified, along with the trematodes *Skrjabinopsolus semiarmatus* and *Skrjabinopsolus nudidorsalis* [37] in *Acipenser ruthenus*, and the monogean trematodes *Diclybothrium armatum* and *Nitzschia storionis*. Of particular zoonotic significance was *Anisakis* sp., a parasite capable of infecting humans and causing anisakidosis [112].

Ibrahimov and Mamedova (2021) [80] identified the cestodes *Bothrimonus fallax* and *Eubothrium acipenserinum* in *Acipenser gueldenstadti* in Azerbaijan. Skóra et al. (2018) [64]

identified nematoda *Raphidascaris acus* in *Acipenser baerii* and *Acipenser ruthenus* in Poland. McCabe (1991) [70] identified the nematoda *Cystinosis acipenseri* in *Acipenser transmontanus* in the USA. Noei et al. (2011) [38] identified six species, from which there were two nematodes, *Cucullanus sphaerocephalus* and *Eustrongylides excisus*, two cestodes, *Amphilina foliacea* and *Bothrimonus fallax*, one trematode, *Skrjabinopsolus semiarmatus*, and one acanthocephalan, *Leptorhynchoides plagicephalus*, in the Caspian sea, Iran.

Nasirov and Bunyatova (2017) [65] identified two nematodes, *Cucullanus* sp. and *Eustrongylides* sp., in Azerbaijan.

Lenhardt et al. (2009) [34] identified the trematoda *Skrjabinopsolus semiarmatus* in *Acipenser ruthenus* in Belgrade, and Foata et al. (2004) [76] identified *Acanthocephala* in the testes of *Acipenser naccarii* in Italy.

Adel et al. (2016) [25] identified the protozoa *Trichodina* sp. and *Ichthyophthirius multifiliis*, the trematode *Diplostomum spathaceum*, the nematodes *Cucullanus sphaerocephalus*, *Anisakis* sp., and *Skrjabinopsolus semiarmatus*, and the Acanthocephala *Leptorhynchoides plagicephalus* in the skin, fin, eyes, and intestines of *Acipenser persicus* in Iran. Atopkin and Shedko (2014) [35] identified the trematode *Crepidostomum oschmarini* in *Acipenser schrenkii* in Russia, and Moghaddam (2013) [26] identified four types of internal helminth parasites, *Cucullanus sphaerocephalus*, *Skrjabinopsolus semiarmatus*, *Eubothrium acipenserinum*, and *Leptorhynchoides plagicephalus* in *Acipenser persicus* in Iran. Rahanandeh et al. (2019) [113] identified the monogenean *Diclybothrium armatum* in the gills of farmed *Huso huso*. Aghaee Moghadam et al. (2014) [114] identified the nematodes *Cucullanus sphaerocephalus* and the Trematoda *Skrjabinopsolus semiarmatus* in the intestines of *Huso huso* in the Caspian sea, Iran, as previously reported by Sattari (2003) [115], who identified *Skrjabinopsolus semiarmatus*, *Leptorhynchoides plagicephalus*, *Cucullanus sphaerocephalus*, *Eubothrium acipenserinum*, *Bothrimonus fallax*, *Eustrongylides excess*, *Anisakis* sp., *Amphilina foliacea*, and *Corynosoma strumosum* in *Acipenser stellatus* in the Caspian Sea, Iran.

*2.3. Copepods*

Small crustaceans called copepods are commonly found in freshwater and marine habitats [42]. The calanoid copepod is the most common form of copepod found in sturgeon fish. It is a tiny planktonic creature that is an important food source for many fish species, including sturgeon. Cyclopoid copepods, Harpacticoid copepods, and Poecilostomatoid are other copepod species that can be found in sturgeon fish. In addition, copepod parasites have been shown to affect the physiological health of the sturgeon and cause anemia. In particular, these ectoparasites can reduce host osmotic competence both directly by damaging and necrosing the epithelium and indirectly by increasing host stress hormone levels [40].

Vasilean et al. (2012) [42] identified the copepoda *Argulus* sp. in *Huso huso* in Romania. The parasite is popularly called "fish lice" and is shaped like a pear, wide at the front and narrow at the end. Its length is about 3.7 mm and it has a pair of tentacles and hooks that are the size of the parasite's body. These crustaceans have bodies adapted to parasitic life in general. Andres et al. (2019) [43] found the copepoda *Argulus flavescens* in *Acipenser oxyrinchus* in the Pascagoula river, USA. The *Argulus flavescens* attaches itself to the gills and skin using its sharp claws and proboscis, causing irritation and inflammation of the skin and feeding on the blood, which can lead to anemia if the infestation is severe [116].

*Ergasilus sieboldi*, *Paraergasilus rylovi*, *Lernaea cyprinacea*, *L. elegans*, *Caligus lacustris* and *Argulus foliaceus* are also copepods that occur on different fishes and are well known to be pathogenic to fishes in aquaculture. Three species are specific to sturgeons, but only one of them, *Pseudotracheliastes stellatus*, is pathogenic. A rather high infection of *A. stellatus* and *A. gueldenstaedtii* by *Pseudotracheliastes stellatus* was noted in the Azov sea. Infection by these species results in quantitative and qualitative changes in white and red blood cells, as diseased fish present anemia [117].

Bauman et al. (2011) [47] identified the copepoda *Argulus* sp. on the gills and skin of *Acipenser fulvescens* in the Marys river, USA. Brown (2010) [48] identified the copepoda

*Dichelesthium oblongum* on the gills of *Acipenser oxyrinchus oxyrinchus* in New York. Munroe et al. (2011) [49] identified the copepoda *Caligus elongatus* and *Dichelesthiumoblongum*, the Hirudinea *Calliobdella vivida*, the Crustacea *Argulus stizostethii*, and the Monogenea *Nitzschia sturionis* in *Acipenser oxyrinchus* in Canada. Bozorgnia (2018) [118] identified the Copepoda *Lernaea cyprinacea* in the gills of *Acipenser stellatus* in the Caspian sea, Iran.

## 2.4. Hirudinea and Polypodiozoa

The Hirudinea *Caspiobdella fadejewi* was identified in wild *Acipenser oxyrinchus* specimens in the Drwêca river, Poland, by Bielecki et al. (2011) [49]. *Caspiobdella fadejewi* can cause physical harm to fish, particularly sturgeon. These leeches attach themselves to the skin of sturgeons and feed on their blood, which can lead to irritation, inflammation, and tissue damage. In severe cases, a large number of leeches can weaken the fish, making it more vulnerable to other predators or diseases. Additionally, leeches can transmit diseases or parasites to the fish, as they can carry a range of harmful pathogens [119]. Bolotov et al. (2022) [60] identified the Hirudinea *Acipenserobdella volgensis* in the pectoral fin of *Acipenseridae* in the Volga river basin in Russia. This species was also found on *Acipenser baerii*, *A. gueldenstaedtii*, and *A. nudiventris* [120].

Raikova (2002) [50] identified the polypodiozoa *Polypodium hydriforme* in *Acipenseriformes* in the Volga river as well. This is the only cnidarian species adapted to intracellular parasitism in fish oocytes. It is a diploblastic animal that possesses stinging cells known as cnidocytes, with a life cycle that consists of two stages: a parasitic stage and a free-living phase. The parasitic stage occurs within host oocytes throughout oogenesis, starting from early previtellogenesis until the hatching stage. The parasite reproduces through longitudinal fission, with the number of tentacles doubling before each division [121]. *Polypodium hydriforme* can affect the reproductive health of the sturgeon in particular, with the infected female sturgeon experiencing reduced egg production and poor egg quality [122,123].

The parasite was also reported by Hoffman et al. (1974) [53] on the eggs of *Acipenser fulvescens* in the USA, as well as by Okamura et al. (2020) [54] and Judd et al. (2022) [56]. Dick et al. (1991) [55] identified the parasite on *Acipenser fulvescens* eggs in Canada. Koshelev et al. (2014) [57] identified *Polypodium hydriforme* on the eggs of *Huso huso dauricus* in Russia, and Mikodina and Ruban (2021) [58] identified them along with the hirudinea *Limnotrachelobdella* in *Acipenser mikadoi*.

## 2.5. Hyperoartia

Hyperoartia, commonly known as lampreys, are parasitic jawless fish that feed by attaching themselves to the body of fish and sucking their blood and body fluids [84]. When lampreys attach themselves to sturgeons, they create open wounds that can become infected and weaken the fish. This makes the fish more susceptible to predation and disease and can also impair their ability to swim and reproduce. Lampreys can also compete with sturgeons for food and habitat, further contributing to population reduction [81]. *Petromyzon marinus* lampreys have been identified by Briggs et al. [81] on *Acipenser fulvescens* in the USA, as well as by Patrick et al. [82], Sepúlveda et al. [83], and Dobiesz et al. (2018) [84]. Almeida et al. (2023) [124], Briggs et al. (2023) [81], and Ionescu et al. (2022) [125] identified the lamprey *Petromyzon marinus* in *Acipenser fulvescens* in Lake Sturgeon, Canada.

## 3. Discussion

According to the literature, sturgeon fish are infected with various ecto- and endoparasite species that live in freshwater, marine areas, lakes, and fish farms. The most commonly reported microhabitats of fish hosts were external organs such as the gills, skin, and the surface of fins. The skin surface, gills, blood, eggs, intestines, gut, and digestive system were the most commonly infected sites (Table 2). Overall, the literature has recognized sturgeon parasite species to include Protozoa, Trematoda, Crustacea, Nematodes, Monogenea, Hirudinea, Copepoda, Acanthocephala, Cestoda, Polypodiozoa, and Hyperoartia (Lamprey).

**Table 2.** Parasites sites of infection.

| Parasites | Infected Organs | | | | | | | | | | | | | | |
|---|---|---|---|---|---|---|---|---|---|---|---|---|---|---|---|
| | **G** | **F** | **S** | **I** | **Eg** | **B** | **BL** | **E** | **GU** | **Ds** | **N** | **SP** | **H** | **T** | **Ca** |
| Protozoa | X | X | X | X | | X | X | X | X | X | X | | | | |
| Monogenea | X | X | X | X | | | | | | | | | | | |
| Acanthocephala | | | X | | | | | | | | | | | X | |
| Trematoda | X | X | X | X | | X | | | | X | | | X | | X |
| Nematodes | X | | X | | | X | | | X | X | | | | | |
| Copepoda | X | X | X | | | X | | | | | | | | | |
| Cestoda | | | | X | | X | | | | | | | | | |
| Polypodiozoa | | | | | X | | | | | | | | | | |
| Hirudinea | X | X | X | | | X | | | | | | | | | |
| Hyperoartia (Lamprey) | | | | | | X | | | | | | | | | |

G gill, S skin, F fin, I intestine, E eye, SP Spleen, H heart, N nose, Eg eggs, B body, BL blood T testes, GU gut, Ga gastrointestinal, Ds digestive system.

By considering all taxonomic groups of parasites (Table 3) that have been the subject of studies concerning the occurrence and absence of parasites in different organs, it is presented in Figure 1 that the intestines are the most susceptible to infestation (30.49%). This is followed by the skin (21.95%), gills (18.29%), and fins (16.46%), while eggs (4.27%), blood (1.83%), and heart (0.61%) are the least susceptible to infestation.

**Table 3.** The taxonomy of parasites hosted by sturgeons.

| Phylum | Class | Order | Family | Species |
|---|---|---|---|---|
| Ciliophora | Oligohymenophorea | Hymenostomatida | Ichthyophthiriidae | *Ichthyophthirius multifiliis* (Fouquet 1876) |
| Ciliophora | Oligohymenophorea | Mobilida | Triochodinidae | *Trichodina* sp. *Trichodina Ehrenberg*, 1830 and *Trichodina* (reticulata Hirschmann & Partsch, 1955) |
| Ciliophora | Oligohymenophorea | Peritrichida | Epistylididae | *Apiosoma* sp. (Blanchard, 1885) |
| Ciliophora | Oligohymenophorea | Sessilida | Epistylididae | *Epistylis* sp. (Ehrenberg, 1830) |
| Platyhelminthes | Trematoda | Diplostomida | Diplostomidae | *Diplostomum spathaceum* (Rudolphi, 1819), Olsson, 1876) |
| Platyhelminthes | Trematoda | Diplostomida | Schistosomatidae | *Schistosoma japonicum* (Katsurada, 1904) |
| Platyhelminthes | Trematoda | Plagiorchiida | Allocreadiidae | *Crepidostomum auriculatum* (Wedl, 1858) Lühe, 1909 |
| Platyhelminthes | Trematoda | Plagiorchiida | Deropristidae | *Pristicola bruchi* (Choudhury, 2009) |
| Platyhelminthes | Trematoda | Diplostomida | Aporocotylidae | *Acipensericola glacialis* (Warren & Bullard, 2017) |
| Platyhelminthes | Trematoda | Plagiorchiida | Deropristidae | *Skrjabinopsolus semiarmatus* (Molin, 1858) Ivanov, 1937 |
| Platyhelminthes | Trematoda | Diplostomata | Aporocotylidae | *Sanguinicola* sp. (Plehn, 1905) |
| Platyhelminthes | Trematoda | Diplostomida | Diplostomidae | *Posthodiplostomum* sp. (Dubois, 1936) |
| Platyhelminthes | Trematoda | Plagiorchiida | Allocreadiidae | *Crepidostomum auritum* (MacCallum, 1919) |

**Table 3.** *Cont.*

| Phylum | Class | Order | Family | Species |
|---|---|---|---|---|
| Platyhelminthes | Trematoda | Plagiorchiida | Allocreadiidae | *Crepidostomum oschmarini* (Zhokhov & Pugacheva, 1998) |
| Platyhelminthes | Trematoda | Plagiorchiida | Deropristidae | *Skrjabinopsolus nudidorsalis* (Ivanov, 1937) |
| Platyhelminthes | Monogenea (Monogenoidea) | Capsalidea | Capsalidae | *Nitzschia sturionis* (Abildgaard, 1794) Krøyer, 1852 |
| Platyhelminthes | Monogenea (Monogenoidea) | Diclybothriidea | Diclybothriidae | *Diclybothriidae gen.* sp. (Bykhovskii and Gusev. 1950) |
| Platyhelminthes | Monogenea (Monogenoidea) | Diclybothriidea | Diclybothriidae | *Diclybothrium* sp. (Leuckart, 1835) |
| Platyhelminthes | Monogenea (Monogenoidea) | Dactylogyridea | Dactylogyridae | *Dactylogyrus* sp. (Diesing, 1850) |
| Platyhelminthes | Monogenea (Monogenoidea) | Diclybothriidea | Diclybothriidae | *Diclybothrium* sp. (Leuckart, 1835) |
| Platyhelminthes | Monogenea (Monogenoidea) | Dactylogyridea | Dactylogyridae | *Dactylogyrus* sp. (Diesing, 1850) |
| Platyhelminthes | Monogenea (Monogenoidea) | Diclybothriidea | Diclybothriidae | *Diclybothrium armatum* (Leuckart, 1835) |
| Platyhelminthes | Monogenea (Monogenoidea) | Diclybothriidea | Diclybothriidae | *Diclybothrium hatum* (Leuckart, 1835) |
| Platyhel mintes | Monogenea | Gyrodactylidea | Gyrodactylidae | Gyrodactylus sp. von Nordmann, 1832 |
| Platyhelminthes | Monogenea | Diclybothriidea | Diclybothriidae | *Paradiclybothrium pacificum* (Bychowsky & Gusev, 1950) |
| Platyhelminthes | Monogenea | Capsalidea | Capsalidae | *Nitzchia* Gervais, 1846 |
| Nematoda | Chromadorea | Rhabiditida | Cystidicolidae | *Capillospirura* sp. (Skrjabin, 1924) |
| Nematoda | Chromadorea | Rhabiditida | Cucullanidae | *Truttaedacnitis* (Cucullanus) (Müller, 1777) |
| Nematoda | Chromadorea | Rhabiditida | Cucullanidae | *Cucullanus sphaerocephlaus* (Rudolphi, 1809) Baylis, 1939 |
| Nematoda | Chromadorea | Rhabiditida | Anisakidae | Anisakis sp. (Dujardin, 1845) |
| Nematoda | Enoplea | Dioctophymatida | Dioctophymatidae | *Eustrongylides excisus* (Jägerskiöld, 1909) |
| Nematoda | Chromadorea | Rhabiditida | Raphidascarididae | *Raphidascaris acus* (Bloch, 1779) Railliet & Henry, 1915 |
| Nematoda | Enoplea | Trichinellida | Cystoopsidae | *Cystoopsis acipenseri* (Wagner, 1867) |
| Nematoda | Chromadorea | Rhabiditida | Anisakidae | *Contracaecum bidentatum* (Ward & Magath, 1917) |
| Nematoda | Chromadorea | Rhabiditida | Anisakidae | *Contracaecum sinipercae* (Dogiel & Achmerov, 1946) |
| Nematoda | Chromadorea | Rhabiditida | Cystidicolidae | *Spinitectus gracilis* Fourment, 1883 |
| Myzozoa | Conoidasida | Eucoccidiorida | Eimeriidae | *Goussia vargai* Cynthia R., Blazer, Vicki S. (2019) |
| Myzozoa | Conoidasida | Eucoccidiorida | Eimeriidae | *Goussia acipensris* (labbe 1896) |
| Myzozoa | Conoidasida | Eucoccidiorida | Haemogregarinidae | *Haemogregarina acipenseris* (Danilewsky, 1885) |

**Table 3.** *Cont.*

| Phylum | Class | Order | Family | Species |
|---|---|---|---|---|
| Euglenozoa | Kinetoplastea | Eubodonida | Cryptobiaceae | *Cryptobia acipenseris* (Joff, Lewashow, Boschenko, 1926) |
| Euglenozoa | Kinetoplastea | Prokinetoplastida | Bodonidae | *Ichthyobodo necatrix* (Henneguy, 1883) |
| Annelida | Clitellata | Rhynchobdellida | Piscicolidae | *Limnotrachelobdella* sp. (Epshtein, 1968) |
| Annelida | Clitellata | Rhynchobdellida | Glossiphoniidae | *Placobdella montifera* (Moore, 1906) |
| Annelida | Clitellata | Rhynchobdellida | Piscicolidae | *Acipenserobdella volgensis* (Epstein, 1969) |
| Annelida | Clitellata | Rhynchobdellida | Piscicolidae | *Piscicola geometra* (Linnaeus, 1761) |
| Arthropoda | Copepoda | Cyclopoida | Ergasilidae | *Ergasilus* sp. (Nordmann, 1832) |
| Arthropoda | Copepoda | Cyclopoida | Lernaeidae | *Lernaea cyprinacea* (Linnaeus, 1758) |
| Arthropoda | Copepoda | Siphonostomatoida | Lernaeopodidae | *Pseudotracheliastes stellatus* (Mayor, 1824) |
| Arthropoda | Ichthyostraca | Arguloida | Argulidae | *Argulus foliaceus* (Linnaeus, 1758) |
| Arthropoda | Ichthyostraca | Arguloida | Argulidae | *Argulus* sp. (Müller O.F., 1785) |
| Arthropoda | Ichthyostraca | Arguloida | Argulidae | *Argulus flavescens* (Wilson C.B., 1916) |
| Arthropoda | Ichthyostraca | Arguloida | Argulidae | *Argulus stizostethii* (Kellicott, 1880) |
| Arthropoda | Copepoda | Siphonostomatoida | Caligidae | *Caligus elongatus* (von Nordmann, 1832) |
| Arthropoda | Copepoda | Siphonostomatoida | Dichelesthiidae | *Dichelesthium oblongum* (Abildgaard, 1794) |
| Arthropoda | Copepoda | Siphonostomatoida | Lernaeopodidae | *Tracheliastes gigas* Richiardi, 1881, Pseudotracheliastes stellatus (Mayor, 1824) |
| Platyhelminthes | Cestoda | Amphilinidea | Amphilinidae | *Amphilina* sp. (Wagener, 1858) |
| Platyhelminthes | Cestoda | Amphilinidea | Amphilinidae | *Amphilina foliacea* (Rudolphi, 1819) Wagener, 1858 |
| Platyhelminthes | Cestoda | Spathebothriidea | Acrobothriidae | *Bothrimonus fallax* (Lühe, 1900) |
| Platyhelminthes | Cestoda | Amphilinidea | Amphilinidae | *Amphilina japonica* (Goto & Ishii, 1936) |
| Platyhelminthes | Cestoda | Bothriocephalidea | Triaenophoridae | *Eubothrium acipenserinum* (Cholodkovsky, 1918) Dogiel & Bychowsky, 1939 |
| Annelida | Clitellata | Rhynchobdellida | Piscicolidae | *Caspiobdella fadejewi* (Epshtein, 1961) |
| Annelida | Clitellata | Rhynchobdellida | Piscicolidae | *Calliobdella vivida* (=Cystobranchus vividus) (Verrill, 1872) |
| Acanthocephala | Palaeacanthocephala | Echinorhynchida | Leptorhynchoididae | *Leptorhynchoides polycristatus* (Amin, Heckmann, Halajian, El-Naggar & Tavakol, 2013) |
| Acanthocephala | Palaeacanthocephala | Echinorhynchida | Paracanthocephalidae | *Acanthocephalus anguillae* (Müller, 1780) |
| Acanthocephala | Palaeacanthocephala | Echinorhynchida | Pomphorhynchidae | *Pomphorhynchus bosniacus* (Kistaroly & Cankovic, 1969) |

**Table 3.** *Cont.*

| Phylum | Class | Order | Family | Species |
|---|---|---|---|---|
| Acanthocephala | Palaeacanthocephala | Echinorhynchida | Leptorhynchoididae | *Leptorhynchoides plagicephalus* (Westrumb, 1821) |
| Chordata | Petromyzonti | Petromyzontiformes | Petromyzontidae | Petromyzon marinus (Linnaeus, 1758) |
| Ciliophora | Phyllopharyngea | Chlamydodontida | Chilodonellidae | *Chilodonella* sp. (Strand, 1928) |
| Cnidaria | Polypodiozoa | Polypodiidea | Polypodiidae | *Polypodium hydriforme* (Ussow, 1887) |

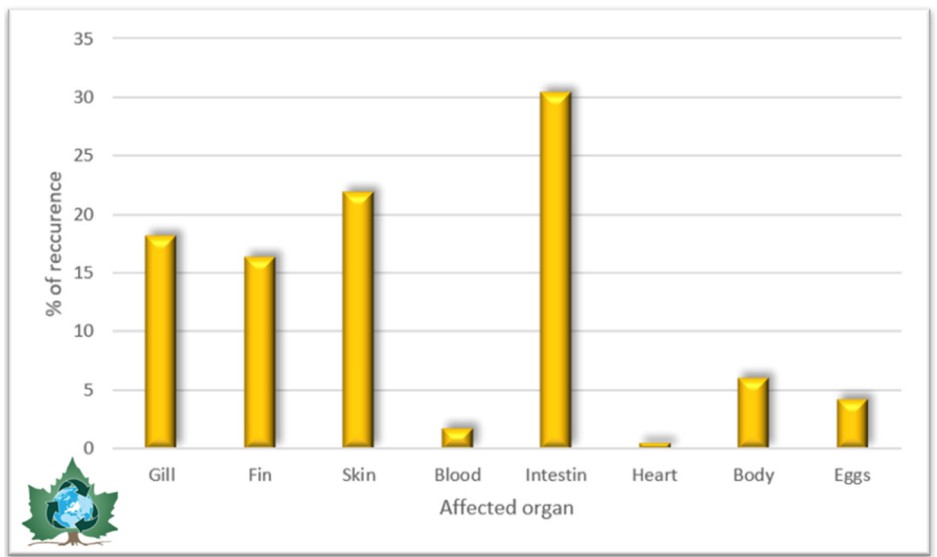

**Figure 1.** Percentage of parasite recurrence by affected organs (all taxonomic groups).

Figure 2 shows that the majority of identifications in the studies have been attributed to protozoa (21%), followed by monogeneans (15%), copepods (14%), cestodes (10%), crustaceans (9%), trematodes (8%), nematodes (7%), hirudines (6%), acanthocephalans (5%), and polypodiozoa (4%). The Hyperoartia group is recorded as having the lowest percentage of identified infestations, at 1%. However, these statistics do not reflect the real in situ scenario, where percentages can vary substantially; rather, they indicate the level of parasite identification based on the literature.

Considering the extent of infestation manifested by various organs of sturgeons in accordance with each taxonomic group of parasites, the scientific literature provides the subsequent data, also presented in the chart in Figure 3:

Protozoa affects the skin the most (48.57%), followed by the fins (22.86%), gills (20%), and blood (8.57%). No species of this group were identified in the intestines, heart, or eggs of sturgeons.

Monogenea affects the skin the most (36%), followed equally by the gills and fins (32%). No species of this group were documented in other organs in sturgeons.

Crustacea affects the skin, gills and fins equally (31.25%), followed by the intestines (6.25%). There were no species of this taxonomic group documented or identified in the blood, heart, or eggs of sturgeons.

Trematoda affects mostly the intestines (92.86%), followed by the heart (7.14%). No species of this group were identified in the gills, fins, skin, blood, or eggs of sturgeons.

Copepoda affects the gills the most (37.5%), followed by the fins and skin equally (25%). The rest of the studies document the infestation of the body as a whole by this group of parasites, while there are no studies about the recurrence of infestation in the blood, intestines, heart, or eggs of sturgeon species.

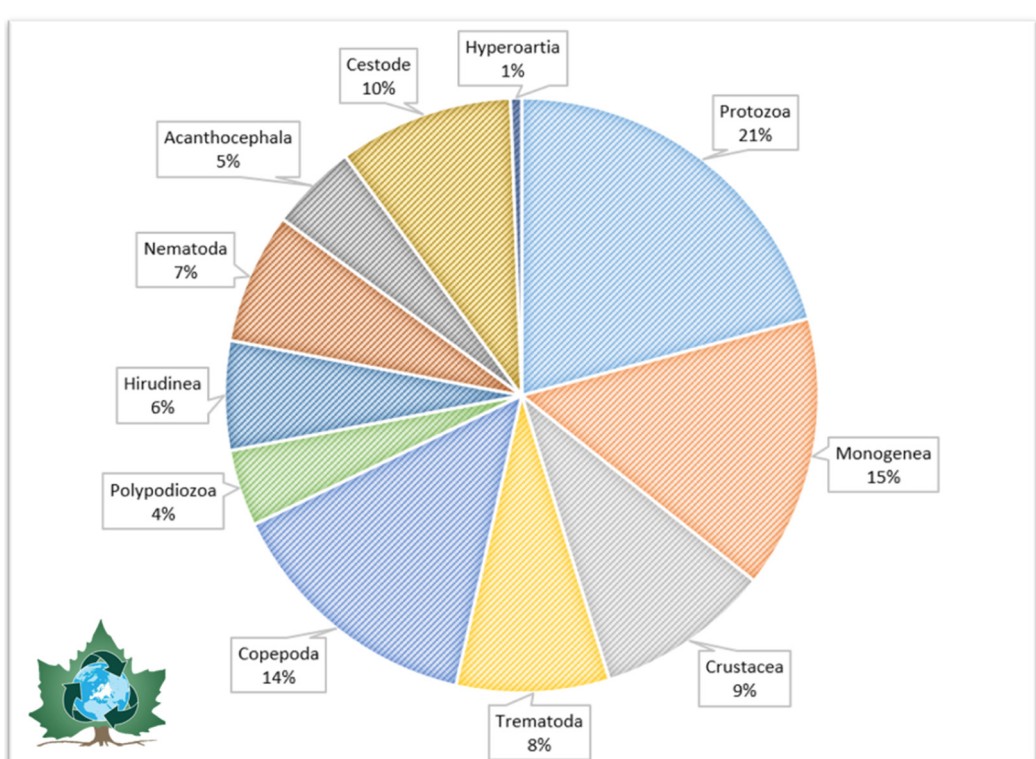

**Figure 2.** Percentage of parasite taxonomic group identification in sturgeon species, based on the reviewed scientific literature.

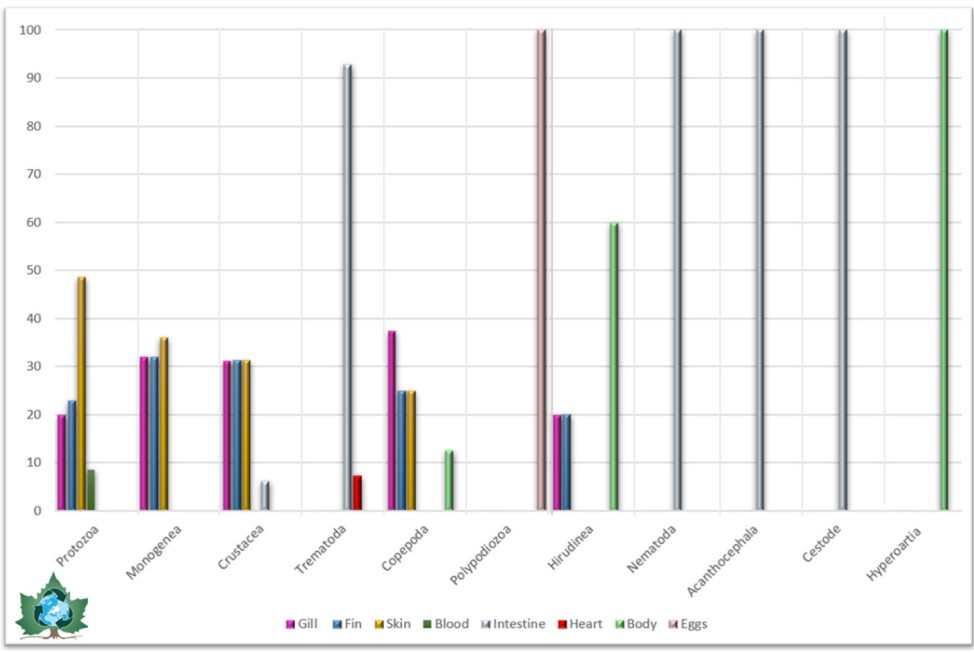

**Figure 3.** Percentage of infestation of each parasite taxonomic group on different organs.

Polypodiozoa species were only identified and documented in the eggs of sturgeons.

Hirudinea species were mostly documented to affect the body as a whole (60%), while 20% of studies identify these parasites in either the gills or fins of sturgeons.

Nematoda species were only identified in the intestines of sturgeon species.

Acanthocephala species were only documented in the intestines of sturgeons.

Cestoda species were only identified in the intestines of sturgeons.

Hyperoartia species were only documented in the body of sturgeons as a whole, with no particular specifications.

## 4. Conclusions

According to the studies, it may be concluded that among the organs of the sturgeon, the intestines are the most prone to parasite infestation, while the blood seems to be the least affected by parasites.

Considering the scientific community's present understanding, most recorded instances of infestation are attributed to protozoa, whereas the group Hyperoartia has the least amount of evidence available.

Furthermore, according to current research, each taxonomic group of parasites exhibits selectivity in terms of the sturgeon organs they target. Nevertheless, the provided statistics are based on the existing level of knowledge, and further research is necessary to gain a more thorough understanding of the impact of each parasite group on sturgeons, which are currently in a critical conservation status globally.

Despite the incomplete understanding of the diversity of parasite species that affect sturgeons, there is less research regarding their ecology, distribution, and prevalence. It is striking that there is currently a lack of scientific focus on understanding the biology and ecology of potentially harmful parasites. Despite a few comprehensive studies on the topic, which have relied solely on the enthusiasm and research efforts of individual scientists, there has been minimal motivation to structure these studies in a more systematic manner.

It is recommended that national and international organizations facilitate more structured research programs regarding fish parasites, especially regarding endangered fish species. These should be designed to allow trend analysis of changes in the parasitic fauna of fishes, such as sturgeons, in the face of environmental changes.

Furthermore, in aquaculture, diseases will continue to play an important role in the economic performance of the industry. Therefore, it is highly recommended that studies be supported and systematically organized to assist in preventing the loss of cultured stock while also preventing aquaculture from becoming a potential reservoir for parasites and disease agents affecting natural stocks.

Time and resources are required to better address the study of parasites that influence sturgeon species from the standpoints of biodiversity monitoring and reducing the risk of disease transmission in natural habitats and aquaculture farms.

**Author Contributions:** Conceptualization, G.D. and A.J.; methodology, A.J. and E.H.; software, I.S.; validation, G.D.; formal analysis, G.D. and I.S.; investigation, A.J.; resources, G.D.; data curation, G.D.; writing—original draft preparation, A.J.; writing—review and editing, A.J. and E.H.; visualization, G.D.; supervision, G.D.; project administration, G.D. All authors have read and agreed to the published version of the manuscript.

**Funding:** This research received no external funding.

**Institutional Review Board Statement:** Not applicable.

**Data Availability Statement:** The data presented in this study are openly available and can be found online by searching all the scientific articles cited.

**Acknowledgments:** The authors would like to thank Daniela VASILE, Furhan T. Mhaisen, Atheer H. Ali and Cristian-Emilian Pop. for their support and considerations.

**Conflicts of Interest:** The authors declare no conflicts of interest.

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
