# Peer review of "Sturgeon Parasites: A Review of Their Diversity and Distribution"

_diversity, doi:10.3390/d16030163_

Round 1
Reviewer 1 Report
Comments and Suggestions for Authors
I would recommend to add the authors names for genus or species names in the tables listing fish and parasite species (see comments in attchment), when introduced for the first time in these tables.

Author Response
Dear reviewer,
Please see attached our reply.
Thank you

Reviewer 2 Report
Comments and Suggestions for Authors
This paper is a review that aims to document “the presence of both ecto- and endo-parasitic species that occur on sturgeon in their natural environments, including lakes, rivers, and controlled systems like aquaculture.” The review is presented in the form of tables, some selected images, and descriptions/discussions of various studies.
A comprehensive review of sturgeon parasites is a challenging task that requires access to a variety of resources, knowledge of the taxonomy/systematics of the parasites, and meticulous attention to detail. Unfortunately, the manuscript falls short in several important aspects. I have made over 60 comments that are directly embedded in the pdf of the paper; I am uploading that as part of the review. I will not repeat all those comments here but will try and summarize the most important aspects. Please note that I read the entire manuscript but stopped making comments or suggesting edits after a certain point because of all the flaws encountered earlier in the manuscript.
11. There are numerous cases in the tables and in the text, where grossly incorrect classification is attributed to many parasites. Trematodes, acanthocephalans, etc. are called nematodes. Even Goussia, a protistan apicomplexan parasite, is called a nematode. In at least one instance, gill monogeneans are listed as intestinal parasites. The gill monogenean, Diclybothrium, is listed as a cestode. These mistakes show a complete unfamiliarity with the parasites fauna and would by themselves be enough cause to disqualify the manuscript from consideration. Please see comments embedded in the uploaded pdf.
22. Several sturgeon parasites are missing from the tables. See comments in uploaded pdf. This compromises the review.
33. It is unclear how the Table is organized. At the beginning, it seems to be organized by taxon, starting with Protozoa, but then the taxonomic organization falls apart and it is evident that the table is organized by study. If so, there is no pattern to that either; the studies are not listed chronologically or alphabetically. Also, in some cases, parasites are listed collectively from multiple species of sturgeons from one study, which is misleading because not all the parasites occur in all the sturgeon species in the study. So, as it currently stands, Table I is confusing and full of errors.
44. The text is largely a summary description of selected studies, but it is not clear why these studies were highlighted. I suggest choosing studies that report parasites of disease importance. Many sturgeon parasites cause no apparent pathology, and these can be excluded from special mention in the discussion.
55. Table I is presented in the introduction! I do not understand why. It should be part of the actual review that follows the introduction.
66. It is unclear why the figures in the manuscript were chosen for inclusion. Why, for example, is there an image of the side view of the trematode, Skrjabinopsolus? The life cycle of Ichthyophthirius is well-known, so is there a rather crude image of it?
My recommendation would be to clean up Table I and make all the necessary corrections. Then, for the main body of the text, to focus on parasites that are of particular interest, i.e. disease causing potential, ability to affect sturgeon populations etc.

Comments on the Quality of English LanguageOverall, the English is fine. Some minor editing is needed.
Author Response
Dear Reviewer,
Please see attached.
Thank you.

Round 2
Reviewer 2 Report
Comments and Suggestions for Authors
The authors have made a good-faith effort to revise the manuscript as suggested. However, there were still instances of incorrect classification (see highlighted portions of the Table and associated comments embedded into the pdf of the revised version, uploaded with this review). Please check the text to ensure that all classifications and taxonomy are correct.

Author Response
Dear Reviewer,
Thank you for your kind suggestions and observations.
We performed the remaining modifications.
Thank you once again for the opportunity given to improve our work.
Best regards,
The authors